# Hypothalamic–Pituitary Diseases and Erectile Dysfunction

**DOI:** 10.3390/jcm10122551

**Published:** 2021-06-09

**Authors:** Gianmaria Salvio, Marianna Martino, Giulia Giancola, Giorgio Arnaldi, Giancarlo Balercia

**Affiliations:** Division of Endocrinology, Department of Clinical and Molecular Sciences (DISCLIMO), Polytechnic University of Marche, Umberto I Hospital, 60126 Ancona, Italy; marianna.martino@ospedaliriuniti.marche.it (M.M.); g.giancola@pm.univpm.it (G.G.); Giorgio.Arnaldi@ospedaliriuniti.marche.it (G.A.); g.balercia@staff.univpm.it (G.B.)

**Keywords:** erectile dysfunction, hyperprolactinemia, acromegaly, growth hormone deficiency, Cushing’s disease, hypogonadotropic hypogonadism, hypopituitarism

## Abstract

Several hormones contribute to ensure penile erection, a neurovascular phenomenon in which nitric oxide plays a major role. Erectile dysfunction (ED), which is defined as the persistent inability to obtain or maintain penile erection sufficient for a satisfactory sexual performance, may be due to arteriogenic, neurogenic, iatrogenic, but also endocrinological causes. The hypothalamus–pituitary axis plays a central role in the endocrine system and represents a fundamental link between the brain and peripheral glands, including gonads. Therefore, the hormonal production of the hypothalamic–pituitary axis can control various aspects of sexual function and its dysregulation can compromise erectile function. In addition, excess and deficiency of pituitary hormones or metabolic alterations that are associated with some pituitary diseases (e.g., Cushing’s disease and acromegaly, hypopituitarism) can determine the development of ED with different mechanisms. Thus, the present review aimed to explore the relationship between hypothalamic and pituitary diseases based on the most recent clinical and experimental evidence.

## 1. Physiology of Penile Erection and Pathophysiology of Erectile Dysfunction

Penile erection is a vascular phenomenon involving neurogenic, psychogenic, and hormonal mechanisms. Sexual stimulation triggers the release of nitric oxide (NO) by nonadrenergic, noncholinergic (NANC) nerves in penile tissue. NO binds to soluble guanylyl cyclase to increase production of 3′,5′-cyclic guanosine monophosphate (cGMP), which activates protein kinase G (PKG) to form a complex cGMP/PKG, inducing relaxation of smooth muscle in the corpora cavernosa and, consequently, dilatation of the cavernous arteries that lead to penile erection [1].

All this, however, occurs thanks to the permissive role of androgens (mainly represented by testosterone, T) that regulate sexual behavior and male reproductive function in numerous ways. First of all, they influence the development of the male reproductive tract in early life and affect sexual behavior and libido in the adulthood [2]: sexual complaints are the most specific symptoms of T deficiency (hypogonadism) in adults [3] and can be reversed via T therapy [4]. T upregulates the activity of the enzyme NO synthase (NOS) by endothelial cells (eNOS) and NANC nerves (nNOS) and downregulates the activity of RhoA-ROCK (Ras homolog gene family member A-Rho-associated, coiled coil containing protein kinase), which is involved in the sensitization to calcium of penile smooth muscle cells, leading in both cases to vasodilation of the penile arteries [3].

When one of the mechanisms described above is compromised, erectile dysfunction (ED), defined as the persistent inability to obtain or maintain penile erection sufficient for a satisfactory sexual performance [5], occurs. ED has a multifactorial etiology with vascular (due to arterial insufficiency or venous incompetency), neurogenic, endocrinological, or iatrogenic causes, but in most cases, it is determined by impaired penile blood flow due to atherosclerosis. As known, many cardiovascular risk factors are associated with ED, and there is a strong association between ED and the development of cardiovascular disease (CVD) [6]. The damage of endothelial cells, caused by numerous conditions such as hypertension, smoking, and diabetes, results in a reduction of NO released in the corpora cavernosa. In addition, low NOS activity is linked to ED, and it has been observed in conditions such as hypercholesterolemia, diabetes, and advanced age [7] as well as in hypogonadism [2], which is associated with increased cardiovascular mortality [8]. In this regard, several studies have documented an age-dependent reduction in circulating T levels in men (a condition named “Late-onset hypogonadism”—LOH) associated with sexual dysfunction and metabolic syndrome [9] and a recent meta-analysis confirmed that low T levels are a marker of cardiovascular risk in aging males [10]. In addition, chronic conditions such as liver cirrhosis, obesity, and hyperinsulinism may increase levels of sex hormone binding globulin (SHBG), which binds circulating T and limits its biological effects [3]. Despite this, the beneficial effects of T therapy on cardiovascular risk are still debated [11], while those on erectile function in patients with hypogonadism are widely recognized [4].

Apart from the known effects of T, little is known about the clinical effects of hypothalamic–pituitary diseases on erectile function, but much evidence suggests that the maintenance of good gonadal and sexual function depends on the proper functioning of a complex hormonal system that intersects both peripherally and centrally. The objective of the present review, therefore, was to examine the most recent experimental and clinical data to provide a holistic view of the endocrine system involving the hypothalamus and pituitary gland as regulatory centers of sexual and erectile function.

## 2. Gonadal Axis (Luteinizing Hormone and Follicle-Stimulating Hormone)

Given the central role of T in driving sexual instinct and maintaining erectile function, gonadotropin deficiency (hypogonadotropic or central hypogonadism) is the major pituitary dysfunction involved in the development of sexual disorders. According to the most recent Endocrine Society guidelines, hypogonadism is a clinical syndrome that results from failure of the testes to produce physiological concentrations of T due to testicular (primary hypogonadism) or hypothalamic–pituitary (secondary hypogonadism) defects. The latter condition is present when T levels are low and gonadotropin levels are low or inappropriately normal [12]. In Europe, hypogonadism affects more than one in 10 people in the 40–79 age group and is classified as hypogonadotropic in 85% of cases [13]. Most cases of hypogonadism in middle-aged and older men are considered to be of functional origin, i.e., due to the functional hypothalamic–pituitary–gonadal (HPG) axis suppression caused by excessive adiposity, comorbid illness, medications, and aging [14]. Like other hormones regulated by the hypothalamic–pituitary axis (e.g., cortisol), T levels show a daily pattern, with the highest values reached in the early morning [15]. The circadian rhythm disruption observed in night-shift workers with sleep disorders is associated with low T levels and hypogonadal symptoms [16], supporting a potential relationship between sleep and sexual function, as further suggested by low serum melatonin levels observed in subjects with ED [17].

Functional hypogonadism (also referred to as LOH) is defined as the coexistence of androgen deficiency-like features and low serum T concentrations in the absence of intrinsic structural HPG pathology in middle-aged and older men, and it is a potentially reversible condition [18]. Additionally, organic hypogonadotropic hypogonadism can result from many conditions and drugs, such as traumatic brain injury (TBI), sellar or suprasellar tumors, iron overload, infiltrative or destructive disease of the pituitary or the hypothalamus, hyperprolactinemia, opioids, narcotics, or glucocorticoids [13].

Drugs may act on gonadal axes in different ways. Use of exogenous androgens, for example, causes feedback suppression of the HPG axis, which may result in hypogonadotropic hypogonadism. This condition, which is easily reversible when caused by short-acting preparations, is associated with a profound suppression of the HPG axis when long-acting drugs or anabolic androgens are used, up to the so-called anabolic steroid-induced hypogonadism [19]. On the other hand, clomiphene and anastrozole have been proposed in the past for the treatment of central hypogonadism. The former is a selective estrogen receptor modulator acting at the pituitary level, while the latter is an inhibitor of the aromatase enzyme, which converts circulating T to estradiol. Both stimulate the release of gonadotropins by the pituitary gland by inhibition of the normal negative feedback exerted by estradiol, with a consequent increase in T levels [20].

Pituitary adenomas are an incidental finding in up to 10% of unselected subjects undergoing a magnetic resonance imaging scan of the brain and in about two-thirds of cases they can be associated with the secretion of excess hormones, such as prolactin (PRL) (32–66%), growth hormone (GH) (8–16%), adrenocorticotropic hormone (ACTH) (2–6%), and, rarely, thyroid-stimulating hormone (TSH) (less than 1%) [21]. Moreover, a significant proportion of these patients (up to 40%), especially those who are older and with macroadenoma, presents a deficiency of one or more pituitary hormones at diagnosis. The gonadal axis was the most frequently affected. As discussed later, hyperprolactinemia, acromegaly, and hypercortisolism are directly associated with altered gonadotropin secretion, often leading to secondary hypogonadism. Nonfunctioning adenomas often stain for gonadotropins or their subunits and occasionally for GH, ACTH, and PRL, but they do not secrete hormones in excessive amounts and are also referred as clinically silent adenomas [21]; when large tumors arise from the pituitary, however, they may cause compression on gonadotropic cells or pituitary stalk deviation, causing hypogonadism either directly or through subsequent hyperprolactinemia, respectively [22].

Clinically functional gonadotroph pituitary tumors are very rare and are morphologically identical to nonfunctioning gonadotroph tumors. Males affected present manifestations of hypogonadism with sexual dysfunction and bilaterally enlarged testicles due to the trophic effect on the testicles with increased length of the seminiferous tubules. An increased sperm count has been rarely reported. In some cases, sexual and erectile dysfunctions may be masked. In children an isosexual precocious puberty has been described, and in one 4-year-old boy with macroadenoma this was associated with galactorrhea, episodic ejaculation, and visual deterioration as well as elevated FSH, LH, T, estradiol, and PRL levels.

Moreover, transsphenoidal surgery and pituitary radiotherapy can result in partial or total hypopituitarism, with subsequent negative effects on sexual function. Postsurgical hypogonadotropic hypogonadism is directly linked to ED and can be significantly reversed by T replacement therapy [23], but other pituitary deficiencies could also play a role. Finally, a possible cause of pituitary damage that should be taken into account is represented by the long-term sequelae of TBIs. Hypopituitarism, indeed, occurs in 15–90% of TBIs with GH deficiency (GHD) (2–66%), adrenal insufficiency (0–60%), hypothyroidism (0–29%), hypogonadism (0–29%), and hyperprolactinemia (0–48%) [24]. After a 10-year follow-up, in a population-based study using The National Health Insurance Research Database of Taiwan, patients with a TBI had a 2.5-fold increased risk of developing ED, whose origin could be caused by a deficiency of one or more pituitary hormones [25].

Although T is the main hormone in the physiological regulation of erectile function, some data suggest that hypothalamic–pituitary hormones may also play a relevant role in sexual function (Figure 1). The effects of individual pituitary hormones on male sexual function and their possible roles in erectile dysfunction are discussed below.

### 2.1. Prolactin

Hyperprolactinemia is the most widely recognized pituitary hyperfunction leading to the development of male sexual dysfunction, although the underlying mechanism is not fully understood. The regulation of PRL release occurs primarily in the hypothalamus, whose dopaminergic neurons inhibit the activity of lactotroph cells. Pituitary stalk deviation by sellar lesions (e.g., pituitary macroadenomas) or drugs that act on the dopaminergic pathway are a common cause of hyperprolactinemia. Furthermore, a large number of medications, such as antipsychotics (phenotiazines, thioxanthenes, butyrophenones, atypical antipsychotics), antidepressants (tricyclic antidepressants, monoamine oxidase inhibitors, selective serotonin reuptake inhibitors), opiates, cocaine, some antihypertensive or gastrointestinal medications (e.g., metoclopramide), as well as estrogens, can cause hyperprolactinemia, mainly via their action on dopamine D2 receptors located in the hypothalamic tuberoinfundibular system and on lactotroph cells [26]. Moreover, PRL-secreting tumors (prolactinomas) arising from the pituitary gland comprise almost 50% of all pituitary adenomas in most series [21]. Hyperprolactinemia is a rare cause of ED (1–5%), but ED and low libido are the most frequent onset symptoms of a prolactinoma in male patients [27], often associated with low serum T levels. Hyperprolactinemia, indeed, affects the HPG axis by decreasing gonadotropin-releasing hormone (GnRH) -mediated stimulation of GnRH receptors, thus disrupting GnRH pulsatility and gonadotropin secretion, inducing a secondary hypogonadism that could explain, at least in part, the sexual symptoms [28]. This is true especially for ED, as suggested by Corona et al., who did not find a significant association between severe hyperprolactinemia and severe ED after adjustment for T levels [29]. On the other side, a direct effect of severe hyperprolactinemia in reducing libido has been demonstrated by some authors [30,31] and may be explained by the presence of PRL receptors in the diencephalic incertohypothalamic dopaminergic system, the most important area for the control of motivational and consummatory aspects of sexual behavior [32]. The normalization of serum PRL levels is therefore recommended for restoration of male sexual desire in patients with hyperprolactinemia and dopamine agonists (bromocriptine and cabergoline) are the first choice, especially for prolactinomas [32]. In males with microprolactinomas receiving dopamine agonists, PRL levels are normalized in 80% of cases and tumors undergo shrinkage or disappear in almost 100% and 45% of cases, respectively [33]. In a recent study, cabergoline, but not bromocriptine, improved erectile and orgasmic function in men with hyperprolactinemia [34]. If T levels are not normalized by dopamine agonists, T supplementation can be added, whereas T therapy alone should be avoided due to the risk of further growth of the adenoma through the aromatization of T in estradiol [22]. For drug-induced hyperprolactinemia, the first choice would be to withdraw or replace the drug involved [26], but this is not always possible, especially for antipsychotics [22]. In patients with risperidone-induced hyperprolactinemia, the administration of adjunctive aripiprazole has shown to be an effective treatment option for normalization of serum PRL levels with no changes in psychiatric or extrapyramidal symptoms [35].

Finally, macroprolactinemia deserves a brief mention. Macroprolactin is a large-sized complex (>150 kDa) resulting from the aggregation of monomeric PRL (~22 kDa) into the bloodstream. If macroprolactin levels increase more than 50–60%, the condition is called macroprolactinemia [36]. Although the majority of authors suggest the use of polyethylene glycol (PEG) precipitation to differentiate between macroprolactinemia and true hyperprolactinemia, as only the latter is considered clinically relevant [22]. Kalsi et al. have recently reported a high percentage (72.73%) of reproductive manifestations related to high PRL levels in 22 patients of both genders with macroprolactinemia. By the way, in the same series, no men with macroprolactinemia complained about ED or low libido, whereas 5% of 80 men with true hyperprolactinemia reported sexual disorders [36]. However, given the small study cohort, it is not possible to draw definitive conclusions and more data are needed to better define this aspect.

Moreover, low PRL levels could play a role in the development of ED, too, as suggested by some authors [37,38], but data in the literature are scarce.

### 2.2. Growth Hormone

During childhood, sex steroids, together with GH and insulin-like growth factor 1 (IGF-1), guide pubertal development, acting in synchronicity: at pre- and peripubertal ages, the administration of exogenous T, both at physiological and pharmacological doses, induces a marked increase in GH secretion. But a similar effect is also observed in elderly and hypogonadic subjects in which an increase in GH levels can be detected in response to the administration of T or GnRH [39]. A further suggestion for a close link between gonadal and somatroph axes has also been detected both at pituitary and testicular levels: pituitary cells expressing mRNA for follicle-stimulating hormone (FSH) and luteinizing hormone (LH) or receptors for GnRH showed a positivity in immunohistochemistry for GH, suggesting a possible mixed function of these cells (somatotropic and, transiently, gonadotropic) or a coregulation of two different hormonal axes by the same population of pituitary cells; on the other hand, the existence of an intra-testicular production of IGF-1 by Sertoli and Leydig cells, which would contribute to spermatogenesis and endocrine function of the testicles, has been observed both in animals and humans [40]. Moreover, GH could exert a direct effect on erectile function. Indeed, a rise in serum GH levels in the cavernous and systemic blood cavities has been observed during the penile stages of flaccidity and tumescence in healthy men but not in subjects with an organic cause of ED, suggesting that disturbances in GH secretion could lead to impaired erectile function [41]. This hypothesis is further supported by experimental data showing that administration of exogenous GH enhances the regeneration of NOS-containing nerve fibers in intracavernosal and dorsal penile nerves after experimentally induced neurotomy [42] and by the clinical finding of increased systemic levels of NO metabolites in GH-deficient patients after treatment with recombinant human GH (rhGH) [43]. Moreover, recent evidence showed that GH plays a positive role in the homeostasis of the vascular endothelium, increasing NO production and protecting the endothelium by a modulation of oxidative stress [44].

#### 2.2.1. Acromegaly

The prevalence of ED is particularly high in patients with acromegaly, in whom it reaches 60% [45]. The association between acromegaly and ED is far from being fully understood, but several hypotheses have been advanced. Data collected using the Structured Interview on Erectile Dysfunction (SIEDY) for pathogenic quantification of ED highlighted the organic component as pathogenetically dominant in the genesis of the dysfunction [46]. Hypogonadism and metabolic complications of acromegaly (diabetes, hypertension, cardiomyopathy, and obstructive sleep apnea syndrome-OSAS) may act in this regard, leading those subjects toward endothelial dysfunction [47]. This was initially suggested by Lotti et al. who analyzed a group of 57 acromegalics, comparing the different characteristics of subjects with (*n* = 24) and without ED (*n* = 33). A greater prevalence of metabolic complications was found among those with ED, with a subsequent increased incidence of major cardiovascular events. A further comparison was made between acromegalic patients with ED and “healthy” subjects with ED. Subjects with acromegaly had a significantly worse ED, a higher organic component of ED according to SIEDY, more prior major cardiovascular events, and worse results at penile doppler ultrasound evaluations [46]. In line with these findings, a recent study showed that GH levels were directly correlated to the severity of ED in 51 males with acromegaly and that NO levels were significantly lower in acromegalic patients than in controls [45], supporting the hypothesis of acromegaly as a direct cause of endothelial dysfunction, as previously suggested by other authors [48]. Moreover, Chen et al. reported higher random GH levels and higher GH nadir after oral glucose tolerance tests (OGTT) in patients with ED, but, surprisingly, no significant association was found between ED and T, body mass index, diabetes, hypertension, or previous coronary artery disease [45].

Taken together, these data clearly indicate that ED is a common manifestation in patients with acromegaly, in whom it occurs more severely than in the general population. GH excess per se could represent a key element in the pathogenesis of this disorder, but little evidence is available. In addition, a possible beneficial effect of controlling acromegaly on erectile function has yet to be evaluated.

#### 2.2.2. Growth Hormone Deficiency

As far as GHD concerns, a recent prospective study found an extremely high prevalence of sexual dysfunctions (71.2%) in 83 adult GHD patients of both sexes. ED had an overall 60% prevalence in men and a 75% prevalence in untreated patients, whereas recombinant GH therapy was associated with a lower, but still relevant, prevalence (35%). It should be noted that the age of men in the untreated group was significantly higher (61.0 ± 12.6 versus 48.1 ± 13.4, *p* = 0.002), even if age and, surprisingly, serum T, were not identified as significant predictors of IIEF-15 scores after logistic regression analysis. A direct correlation between serum IGF-1 and IIEF-15 scores was also evident, but a direct effect of GH or IGF-1 on erectile function could not be clearly distinguished by the effect of cardiovascular changes related to GHD, as discussed by the authors [49].

### 2.3. Adrenocorticotropic Hormone

ACTH as other neuropeptides (dopamine, serotonin, noradrenaline) may have a role in central regulation of the erectile process. In rats, the administration of ACTH induced penile erection, most probably mediated via stimulation of melanocortin receptors [50]. Briefly, in these animals, ACTH–melanocyte-stimulating hormone (MSH) peptides facilitated penile erection, ejaculation, and copulatory behavior [51].

#### 2.3.1. Cushing’s Disease

A possible link between glucocorticoids and ED has also been hypothesized. With the initiation of penile erection, mean serum cortisol levels decline in the systemic circulation and in the cavernous compartment, suggesting that a drop in systemic cortisol levels could represent a prerequisite to enable an adequate genital response to sexual stimulation [52]. These data are further supported by the evidence of high serum cortisol levels in subjects with stress-related impairment of the response to intracavernosal injection of smooth muscle relaxants [53]. In patients with chronic diseases, the administration of exogenous corticosteroids at supraphysiological doses is associated with the development of ED [54]. Moreover, up to 69% of men with endogenous hypercortisolism complain about decreased libido, maybe due to associated conditions such as depression and hypogonadism, but the exact prevalence of ED in these patients is unknown [55].

In the early 1970s, Gabrilove et al. wrote “reports on testicular function are rare in patients with Cushing’s syndrome”. Almost 50 years later, data on sexual and reproductive function in individuals with this condition are still surprisingly scarce. Milcou and Stoica were the first authors who reported loss of libido, impotence, and difficulty in achieving orgasm in male subjects with Cushing’s disease [56]. The first histological postmortem findings showing tubular atrophy and absence of Leydig cells in the testes of 2 patients [57] were subsequently confirmed by the examination of 4 subjects with active Cushing’s syndrome that underwent histologic examination of the testes, showing different patterns of disorganization of the seminal epithelium and decreased numbers of Leydig cells [56]. Some years later, Luton et al. was the first to report that hypogonadotropic hypogonadism is a common finding in males with active Cushing’s disease and that it is reversible when hypercortisolism is corrected [58].

There is also indirect evidence linking forms of functional hypercortisolism (also referred to as pseudo-Cushing’s) to the development of ED. In subjects without pituitary or adrenal disorders, cortisol levels, measured both in serum and in saliva, have shown significant inverse correlations with erectile function, sexual desire, short version of International Index of Erectile Function (IIEF) score, and total IIEF score [59]. Moreover, serum cortisol levels and T levels showed inverse correlation in patients with sleep disorders and major depression [60]. Chronic conditions such as major depression, alcoholism, diabetes mellitus, and obesity, indeed, can activate the hypothalamic–pituitary–adrenal axis, resulting in a state of mild clinical and biochemical hypercortisolism that could lead to the same metabolic and cardiovascular complications of Cushing’s syndrome [61]. In a pilot study, Kalaitzidou et al. enrolled 31 men newly diagnosed with ED to investigate the effects of a tadalafil plus an 8-week stress management program (*n* = 19) versus tadalafil alone. Although no significant differences were noted in morning cortisol levels between the two groups either before or after treatment, after 8 weeks of interventions a negative correlation was found between erectile function and perceived stress score (ρ −0.407, *p* = 0.023) and between the increase in cortisol levels 45 min after awakening (i.e., cortisol awakening response) and IIEF-15 sexual desire score (ρ −0.402, *p* = 0.025), suggesting that the negative effect of stress on libido could be mediated by serum cortisol levels [62]. In this regard, the negative effect of the hypothalamic–pituitary–adrenal axis hyperactivation on erectile function was confirmed by demonstrating that among subjects with ED there is a higher rate of non-suppressors to dexamethasone administration than in both subjects with depression and healthy individuals [63]. Moreover, functional hypercortisolism can worsen sexual function in subjects with diabetes mellitus and LOH, two frequently associated conditions [64], and in patients with OSAS [65].For this purpose, it should be remembered the wide overlapping between the clinical features of hypercortisolism (both functional and pathological) and those of metabolic syndrome, such as diabetes mellitus, hypertension, obesity, and insulin resistance [66], which are strongly correlated with the development of vasculogenic ED [6]. Therefore, a multifactorial approach is essential for patients with this problem.

#### 2.3.2. Adrenal Insufficiency

Little is known about the positive effect of cortisol on erectile function and the sexual effects of hypocortisolism, but animal models suggest that corticosteroids may act in concert with T in maintaining nNOS activity at the penile level [67]. This seems to be confirmed in a study on subjects with Addison’s disease in whom short-term hormone replacement treatment resulted in a dramatic improvement in sexual function (including erectile function) [68], but no conclusions about central hypocortisolism can be drawn from this since the possible effect of mineralocorticoid substitution must also be taken into account.

### 2.4. Thyroid-Stimulating Hormone

Since normal thyroid function is important for maintaining normal reproductive and sexual functions, it is not surprising that patients with hypothyroidism and hyperthyroidism may have some degree of impaired sexual function. Thyroid hormones, indeed, lower the bioavailability of T, increasing the concentration of SHBG, which in turn binds circulating T and hampers its effect on tissues. In addition, since thyrotropin-releasing hormone (TRH) acts as a stimulating factor for PRL, prolonged primary hypothyroidism may result in a stimulus on the hypothalamic–pituitary–thyroid axis with consequent hyperprolactinemia. Nevertheless, it is still unclear to what extent these disorders can contribute to ED [69]. Genetically increased or decreased TSH levels (within the normal range) are inversely associated with SHBG concentrations but not with sexual disfunction [70]. Only overt hypo- or hyperthyroidism are indeed associated with sexual dysfunction, which can be resolved with the return to the euthyroid state [71]. Therefore, beyond the effects of thyroid hormones, whether TSH may have a direct effect on male sexual function has yet to be determined.

### 2.5. Vasopressin and Oxytocin

Vasopressin and oxytocin are neuropeptides synthesized in the hypothalamus and secreted from the posterior pituitary gland. Vasopressin plays an essential role in the control of water and osmotic balance. Oxytocin is involved in parturition and lactation. In addition to these physiologic functions, both neuropeptides are involved in numerous activities and behaviors in mammals. In particular, oxytocin has important metabolic effects and has a role in physical and mental protective adaptations and sexual emotions lending the name of “love hormone”. In men, plasma oxytocin levels increase markedly after ejaculation. In addition, oxytocin injected into the spine of male rats causes increased sexual activity and neuronal activity in animals. However, mechanisms involving oxytocin in male sexual function and behavior are not clearly known [72]. Similarly, the role of vasopressin in sexual function needs to be clarified. Given that patients with diabetes insipidus often have hypopituitarism and hypogonadism, it is difficult to distinguish the effects of hypogonadism from those of vasopressin deficiency.

Characteristics of hypothalamic–pituitary diseases associated with ED and proposed mechanisms are summarized in Table 1.

## 3. Conclusions

Although evidence suggests an important role for hypothalamic–pituitary hormones in the control of erectile function, the actual incidence of ED in pituitary disorders is poorly understood. Since ED presents an important effect on quality of life, as well as representing an early sign of endothelial dysfunction, it should always be investigated, preferably using validated questionnaires (e.g., IIEF-15 and SIEDY), and treated. In addition to the use of drugs with proven benefit on erectile function such as phosphodiesterase-5 inhibitors, the correction of all cardiovascular risk factors and the restoration of normal pituitary function appear essential in these patients.

## Figures and Tables

**Figure 1 jcm-10-02551-f001:**
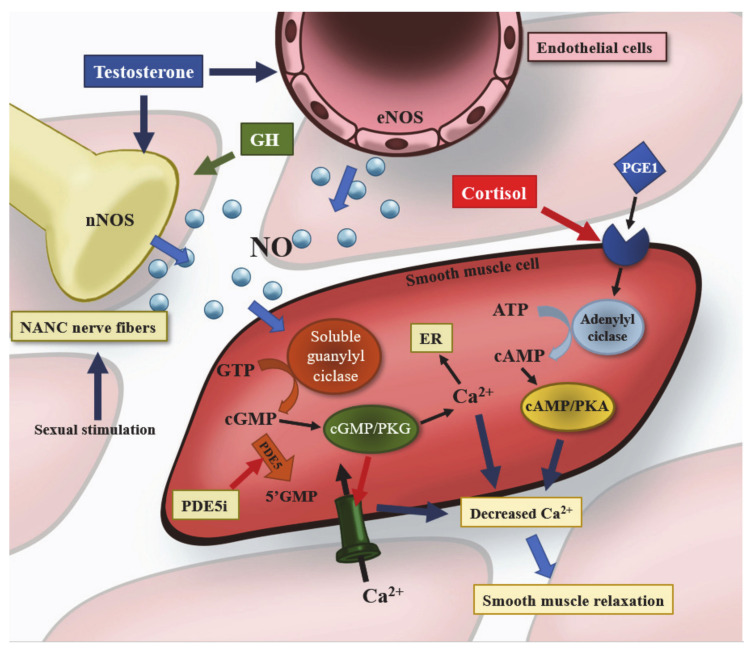
Mechanism of penile erection and hormonal interactions. Nitric oxide (NO) is produced by the enzyme NO synthase (NOS), which is located inside nonadrenergic noncholinergic (NANC) nerves (nNOS) and endothelial cells (eNOS). Sexual stimulation triggers the release of NO by the NANC nerves, which is enhanced by growth hormone (GH) and testosterone. The latter also upregulates the activity of NOS in both nerves and endothelial cells. In penile smooth muscle cells, NO binds to soluble guanylyl ciclase to increase production of 3′,5′-cyclic guanosine monophosphate (cGMP) from guanosine-5′-triphosphate (GTP), which activates protein kinase G (PKG) to form a complex cGMP/PKG, which in turn inhibits the entry of extracellular calcium and promotes the shift of intracellular calcium into the endoplasmic reticulum (ER). Decreased calcium concentration leads to smooth muscle relaxation and, consequently, dilatation of the cavernous arteries and penile erection. Activity of cGMP/PKG is physiologically limited by the enzyme phosphodiesterase 5 (PDE5), which is the target of PDE5 inhibitor drugs (PDE5i), the first line of treatment for erectile dysfunction (ED). Prostaglandin 1 (PGE1), which is administered through intracavernosal injections for the treatment of ED, binds a G-protein-coupled receptor that stimulates adenylyl cyclase activity, which promotes the production of cyclic adenosine monophosphate (cAMP) from adenosine triphosphate (ATP). As for cGMP, cAMP activates a protein kinase (protein kinase A—PKA) and the complex cAMP/PKA reduces intracellular calcium levels and causes smooth muscle relaxation. High serum cortisol levels inhibit the response to intracavernosal injection of smooth muscle relaxants via an unknown mechanism.

**Table 1 jcm-10-02551-t001:** Hypothalamic–pituitary diseases associated with erectile dysfunction and proposed mechanisms.

Dysfunction	Causes	Mechanisms
Pituitary Hormones Excess		
Hyperprolactinemia	Pituitary adenoma (prolactinoma) Sellar/parasellar masses Drugs Stalk effect	- Secondary hypogonadism due to disruption of GnRH pulsatility and gonadotropin secretion - Reduced libido due to direct effect on the central nervous system
Hypercortisolism	Corticotroph adenoma (Cushing’s disease) Functional hypercortisolism	- Secondary hypogonadism due to disruption of GnRH pulsatility and gonadotropin secretion - Primary hypogonadism due to decreased number of Leydig cells in the testes - Endothelial dysfunction due to metabolic comorbidities
Acromegaly	Pituitary adenoma	- Secondary hypogonadism due to disruption of GnRH pulsatility and gonadotropin secretion. - Endothelial dysfunction due to metabolic comorbidities.
Hypopituitarism		
Hypogonadotropic hypogonadism	Pituitary adenoma Sellar/parasellar masses Drugs Functional hypogonadism TBI TNS Radiotherapy	- Reduced libido and dysregulation of sexual behavior - Endothelial dysfunction due to metabolic comorbidities - Reduced NO production in penile tissue due to reduced eNOS and nNOS activity
GHD	TBI TNS Radiotherapy	- Reduced NO production in penile tissue due to reduced nNOS activity
Hypocortisolism	TBI TNS Radiotherapy	- Unknown

GHD = growth hormone deficiency; NO = nitric oxide; TBI = traumatic brain injury; TNS = transsphenoidal surgery.

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
