# Peer review of "Hypothalamic–Pituitary Diseases and Erectile Dysfunction"

_jcm, 2021, doi:10.3390/jcm10122551_

Round 1
Reviewer 1 Report
Congratulation to the authors, great manuscript however, I think the manuscript can be improve.
In the first section you fail to address what urologist call venous leak as the cause of ED.
Please add a section explain the effect of hypothyroidism and hyperthyroidism
Also how clinical status that affects the free hormones affect ED (i.e., liver cirrhosis)
Please explain in length the effect of how medications such as exogenous testosterone (shot and long-acting), bHCG, Clomid, and anastrozole over ED.
Author Response
Congratulation to the authors, great manuscript however, I think the manuscript can be improve.
In the first section you fail to address what urologist call venous leak as the cause of ED.
- Venous leakage has been added among the causes of ED as suggested
Please add a section explain the effect of hypothyroidism and hyperthyroidism
2. A section on thyroid function has been added as suggested.
Also how clinical status that affects the free hormones affect ED (i.e., liver cirrhosis)
3. We have added a paragraph as suggested
Please explain in length the effect of how medications such as exogenous testosterone (shot and long-acting), bHCG, Clomid, and anastrozole over ED.
4. We have added a paragraph as suggested
Thank you for your valuable suggestions. Further minor enhancements have been added to improve the readability of the article.

Reviewer 2 Report
Generally, the review is written satisfactory.
However, I have following suggestions for improvements.
1. It would benefit form more structure. Pituitary hormones and ED (headline 2) is followed by Pituitary hyperfunctioning and ED (headline 3) and Hypopituitarism and ED (headline 4). While most of headline 2 is about GH (27 lines), other hormones receive very little attention (4-8 lines). Headline 3 has subheadings: hyperprolactinemia, Cushing’s disease and acromegaly. Headline 4 is unstructured.
Thus, it would be helpful and improve the reading experience if the authors would arrange the review by hormones, e.g., Somatotropes: growth hormone, Corticotropes: ACTH, Thyrotropes: Thyroid-stimulating hormone (TSH), Gonadotropes: Lactotropes: Prolactin, Oxytocin
or use a similar structure with subheadings to keep better track and ease the reading experience.
Also, always follow the same order of using hyper- and hypo-functioning.
2. TSH should be given more attention, and following papers used and cited:
Gabrielson AT, Sartor RA, Hellstrom WJG. The Impact of Thyroid Disease on Sexual Dysfunction in Men and Women. Sex Med Rev. 2019;7(1):57-70. doi:10.1016/j.sxmr.2018.05.002.
AND
Kjaergaard AD, Marouli E, Papadopoulou A et al. Thyroid function, sex hormones and sexual function: a Mendelian randomization study. Eur J Epidemiol. 2021 Mar;36(3):335-344. doi: 10.1007/s10654-021-00721-z.
3. It would be useful if the autors could provide number of individuals included in the studies cited. Only percentages are given, and providing total N would be helpful in quickly asessing the value of the findings presented.
4. Conclusions should be more concrete. How should ED be investigated and treated?
Minor:
Lines 70-73 starting with ”However,” hold no information, and should be omitted.
By the way, line 113 is a strange way to start this sentence. Did the authors mean however?
Line 284 ”supposed to be” should be replaced by ”considered to be”
Line 289-299 By the other side is wrong. Do authors mean Additionally?
Lines 322 and 325. Two sentences in a row start with anyway. Is anyway indeed what the authors mean, or could a synonym be used?
Round 2
Reviewer 1 Report
I think will be interest to discuss the mechanism of the circadian rhythm of testosterone, and if there is any effect research that support sleeping patterns and ED Is there any report of ED and abnormal levels of melatonin?
Author Response
This is a valuable topic and we therefore included changes in circardian testosterone rhythm among the cuases of functional hypogonadism (lines 82-87). We found only one report about low serum melatonin levels and ED and we were surprised to find that the effect of melatonin supplementation on humans has never been studied. Thanks for this useful observation.